# Long-Term Outcomes of Multidrug-Resistant *Pseudomonas aeruginosa* Bacteriuria: A Retrospective Cohort Study

**DOI:** 10.3390/antibiotics13080685

**Published:** 2024-07-24

**Authors:** Chisook Moon, Jin Suk Kang, Seok Jun Mun, Si-Ho Kim, Yu Mi Wi

**Affiliations:** 1Division of Infectious Diseases, Department of Internal Medicine, Inje University College of Medicine, Busan 47392, Republic of Korea; duomon@hanmail.net (C.M.); gmlsenddl06@naver.com (J.S.K.); kajama37@gmail.com (S.J.M.); 2Division of Infectious Diseases, Samsung Changwon Hospital, Sungkyunkwan University School of Medicine, Changwon 51353, Republic of Korea; wychhazel@naver.com

**Keywords:** *Pseudomonas*, multi-drug resistance, bacteriuria, urinary tract infection, pneumonia, bacteraemia

## Abstract

The relationship between bacteriuria and subsequent symptomatic infections, particularly bacteraemia, has been a subject of ongoing research. We aim to investigate the clinical characteristics, long-term outcomes, and factors associated with subsequent symptomatic infection following an initial multidrug-resistant *P. aeruginosa* (MDRP) bacteriuria episode. A retrospective cohort study was conducted among patients with MDRP bacteriuria who were hospitalized at a tertiary care hospital from 2009 to 2018, with a 12-month follow-up period for each patient. The primary endpoint was the incidence of subsequent symptomatic MDRP infections at any site, and the secondary endpoint was the overall mortality rate. A total of 260 patients with MDRP bacteriuria were included in the analysis, of whom 155 patients (59.6%) had asymptomatic bacteriuria. Subsequent symptomatic MDRP infections were documented in 79 patients (30.3%) within 12 months of the initial bacteriuria episode: UTI (n = 47, 18.1%), pneumonia (n = 21, 8.1%), bacteraemia (n = 9, 3.5%), soft tissue infection (n = 7, 2.7%), and bone and joint infection (n = 4, 1.5%). Intensive care unit (ICU) acquisition and recurrent bacteriuria were independent risk factors of subsequent symptomatic infections in patients with MDRP bacteriuria. The overall mortality rate was 16.9%, with 31.8% of deaths estimated to be associated with MDRP infection. Solid tumours, cardiovascular diseases, chronic liver disease, chronic lung disease, ICU acquisition, absence of pyuria, and concurrent MDRP bacteraemia were independent predictors of mortality. MDRP bacteriuria has the potential for progression to symptomatic infection and associated mortality. Targeted interventions and prevention strategies were crucial to reduce subsequent infections in patients with MDRP bacteriuria, especially in high-risk patients.

## 1. Introduction

Urinary tract infections (UTIs) represent the most prevalent form of nosocomial infections [1], with a growing concern over the increasing incidence of antibiotic-resistant gram-negative pathogens in recent years. Among these, *Pseudomonas aeruginosa* is responsible for a significant proportion of UTI cases, accounting for 7–10% of infections [2]. Of particular concern is the rise of multidrug-resistant *P. aeruginosa* (MDRP) strains, which have become increasingly prevalent in hospital settings [3,4,5]. MDRP poses a significant challenge in healthcare environments due to its ability to transmit within hospitals, potentially triggering outbreaks [6,7,8]. This characteristic, combined with its resistance to multiple antibiotics, complicates treatment strategies and raises concerns about patient outcomes. Although some antibiotics have shown efficacy against MDRP, optimal treatment approaches remain unclear, necessitating a delicate balance between judicious antimicrobial use and the risks associated with disseminated multidrug-resistant infections [9,10,11].

Bacteriuria often leads to unnecessary antimicrobial use, and urinary drainage systems can serve as reservoirs and potential sources of multidrug-resistant bacteria transmission to other patients [12,13]. The relationship between bacteriuria and subsequent symptomatic infections, particularly bacteraemia, has been a subject of ongoing research. Previous studies have identified several risk factors for bacteraemia originating from urinary sources, including diabetes, immunosuppression, catheterization, and the presence of shaking chills [14,15,16]. However, studies investigating the factors associated with subsequent symptomatic infection in patients with MDRP bacteriuria are lacking. Moreover, the long-term outcomes and mortality of patients with MDRP bacteriuria have not yet been defined. Therefore, our study aimed to investigate the clinical characteristics, long-term outcomes, and factors associated with symptomatic infection and mortality among patients with MDRP bacteriuria, and to develop a management strategy for this condition.

## 2. Results

### 2.1. Demographic and Clinical Characteristics

A total of 289 patients with MDRP bacteriuria were identified during the study period. Twenty-nine patients were excluded due to symptomatic infection with MDRP prior to bacteriuria development (n = 9), or unavailability of medical records (n = 20). Of the 260 enrolled patients with MDRP bacteriuria, 59.6% (n = 155), 34.2% (n = 89), and 6.2% (n = 16) were classified as having asymptomatic bacteriuria (ASB), catheter-related UTI, and catheter-free UTI, respectively (Figure 1). The median age at the time of diagnosis was 70 years, with the majority being male patients. At the time of bacteriuria diagnosis, 6 (2.3%) patients had concomitant MDRP bacteraemia. A total of 233 (89.6%) patients had a history of antimicrobial exposure within 90 days, and 184 (70.8%) had neurological diseases such as stroke or spinal cord injury. Recurrent bacteriuria occurred in 90 (34.7%) patients. Among 260 isolates, 69 (26.5%) and 191 (73.5%) were MDRP and extensively drug-resistant Pseudomonas (XDRP), respectively. No pan-drug-resistant strain was detected. During the bacteriuria episode, 70 (37.2%) were prescribed active antibiotics against MDRP bacteriuria. Of 228 patients with catheter-related bacteriuria, catheter removal was performed within 7 days in 38 (16.7%), and catheter exchange was performed in 136 (59.6%), while the catheter was maintained in 37 (16.2%) (Table 1).

### 2.2. Risk Factors for Symptomatic MDRP Infection in Patients with MDRP Bacteriuria

Within 12 months of bacteriuria onset, the following symptomatic MDRP infections were documented in 79 patients (30.3%): UTI (n = 47, 18.1%), pneumonia (n = 21, 8.1%), bacteraemia (n = 9, 3.5%), soft tissue infection (n = 7, 2.7%), and bone and joint infection (n = 4, 1.5%). The median duration from the documentation of MDRP bacteriuria to subsequent symptomatic infection manifestation was 12 days (range: 2–356 days). Univariate analysis showed that ICU acquisition (*p* = 0.04), underlying urological disease (*p* = 0.02), symptomatic bacteriuria at initial episode (*p* = 0.03), active antibiotic therapy for MDRP (*p* = 0.02), and recurrent bacteriuria episodes (*p* < 0.01) were risk factors associated with the development of symptomatic MDRP infection within 12 months. Notably, urinary catheter removal within 7 days had a significantly lower incidence of symptomatic infection than when the catheters were maintained (13.2% vs. 36.4%, *p* < 0.01) (Table 1). Multivariate logistic regression analysis identified ICU acquisition and recurrent bacteriuria episodes as independent risk factors for the development of symptomatic infection within the 12 months (Table 2).

### 2.3. Risk Factors for Overall Mortality in Patients with MDRP Bacteriuria

Within 12 months of documented MDRP bacteriuria, 44 patients (16.9%) died, with a third of deaths estimated to be associated with MDRP infection (n = 14). The median duration between MDRP bacteriuria and death was 20 days, with a range of 2 to 321 days. Table 3 shows the characteristics of patients between survivors and non-survivors in patients with MDRP bacteriuria. Multivariate analyses showed that solid tumours, cardiovascular diseases, chronic liver disease, chronic lung disease, ICU acquisition, absence of pyuria, and concurrent MDRP bacteraemia during the initial bacteriuria were independent factors associated with 12-month overall mortality (Table 4).

## 3. Discussion

Our study showed that MDRP bacteriuria was frequently associated with asymptomatic rather than symptomatic bacteriuria. Approximately 30% of patients developed symptomatic MDRP infections within 12 months of the initial MDRP bacteriuria episode. ICU acquisition and recurrent bacteriuria were independent risk factors for subsequent symptomatic infection. All-cause mortality within 12 months of MDRP bacteriuria occurred in 16.9% of patients, with a third of deaths attributable to MDRP. Underlying solid tumours, cardiovascular diseases, chronic liver disease, chronic lung disease, ICU acquisition, absence of pyuria, and concurrent MDRP bacteraemia were independently associated with mortality. Our findings provide important insights into the clinical course of MDRP bacteriuria and could contribute to the development of targeted interventions and prevention strategies to reduce subsequent infections in this population.

Our study showed that 30.3% of patients with MDRP bacteriuria developed symptomatic infections within 12 months, with a median duration of 12 days between the onset of MDRP bacteriuria and the subsequent symptomatic infection. The incidence of bacteraemia in our study was 3.5%, which falls within the range of 0.4% to 4% reported in previous studies for patients with catheter-related bacteriuria who progress to bacteraemia [17,18,19]. Intriguingly, our study also found that patients with MDRP bacteriuria developed pneumonia (8.1%). ICU acquisition and recurrent bacteriuria episodes were associated with subsequent symptomatic MDRP infection within 12 months. Conway et al. [17] identified several independent risk factors for subsequent bacteraemia in patients with CAB, including younger age, male sex, immunosuppressant use, urologic procedures, non-enterococcal bacteriuria, longer prior hospital stay, and maintaining catheters after bacteriuria onset. Similarly, Bursle et al. [18] reported that catheter insertion in operating rooms, chronic kidney disease, higher age-adjusted Charlson comorbidity index, use of catheter for urine output monitoring, and dementia were independent predictors of subsequent bacteraemia in patients with CAB. Advani et al. [19] also observed that male sex, hypotension, meeting the systemic inflammatory response syndrome criteria, urine retention, fatigue, serum leucocytosis, and pyuria were independent risk factors for subsequent bacteraemia. Notably, our study revealed that the incidence of subsequent infections was lower among patients whose catheters were removed compared to those whose catheters were exchanged or maintained (13.2% vs. 36.4%; *p* < 0.01). This finding is consistent with the results of previous research showing that early catheter removal alleviates the risk of subsequent infections [20,21], and it aligns with Conway et al.’s observations [17] that maintaining catheter after bacteriuria onset independently increases the risk of subsequent bacteraemia. By taking patient-specific factors into account when assessing the risk of bacteraemia, clinicians can avoid unnecessary antibiotic administration in low-risk patients while ensuring prompt treatment in patients at highest risk for complications. The personalised risk-based approach to empiric antibiotic therapy proposed by Advani et al. [19] could provide valuable guidance for more targeted treatment decisions in patients with MDRP bacteriuria.

Our study showed that all-cause mortality rate within 12 months of MDRP bacteriuria was 16.9%, with 31.8% of deaths attributable to MDRP infection. These findings highlight the potential fatality of MDRP bacteriuria within a year of onset, especially among hospitalised patients with underlying chronic diseases. Underlying solid tumours, cardiovascular diseases, chronic liver disease, chronic lung disease, ICU admission, absence of pyuria, and concurrent MDRP bacteraemia during the initial bacteriuria episode were significantly associated with mortality. Notably, concurrent bacteraemia was shown to be an independent factor associated with mortality within 12 months. Given the lack of effective treatment options for MDRP infections, this finding is consistent with the results of previous studies indicating a direct correlation between inappropriate antibiotic treatment and patient survival in bacteraemia cases [6,22]. In our study, we also observed a significantly higher mortality rate within 12 months among patients without pyuria than those with pyuria (26.3% vs. 13.9%, *p* = 0.03). This pattern is similar to the findings of patients with Staphylococcus aureus bacteriuria [23,24]. Bacteriuria without pyuria can originate from sample contamination during urine collection or colonisation within the urinary catheters, but it can also indicate disseminated infection subsequent to bacteraemia [20]. In the case of secondary bacteriuria associated with disseminated infection, the likelihood of inducing a local inflammatory response in the urinary tract may be reduced [23,24]. Consequently, if contamination or colonisation can be reasonably excluded in patients with MDRP bacteriuria, clinicians should be mindful of the possibility of accompanying bacteraemia and require a thorough clinical assessment.

Our study has some limitations. First, it was conducted as a single-centre study and the number of MDRP samples was relatively small, so there may be limitations in generalizing the findings to other hospitals and regions. Second, because of the retrospective nature, data collection was limited by data availability, and patient management lacked standardisation. The classification of UTI and various symptomatic infections relied on available data and may not have reflected epidemiological or clinical trial results from other centres. Third, our study included the lack of data on the duration and pattern of MDRP shedding in patients with bacteriuria. The temporal dynamics of pathogen shedding could potentially influence the risk of subsequent infections and transmission events. This information could provide valuable insights into the optimal duration of infection control measures and inform decisions about repeat cultures in the clinical management of MDRP bacteriuria. Lastly, while our findings identify several significant predictors of subsequent symptomatic infection and mortality, the wide confidence intervals, particularly for variables with low event rates, suggest a degree of uncertainty in the magnitude of these associations. For instance, the small number of patients with ICU admission (n = 21), chronic liver disease (n = 14), chronic lung disease (n = 12), and concurrent bacteraemia (n = 6) contribute to the imprecision of these estimates. These results should be interpreted as indicative of potential risk factors that warrant further investigation in larger, prospectively designed studies with a priori sample size calculations to ensure adequate statistical power for less common predictor variables. Notwithstanding these limitations, our data suggest that MDRP bacteriuria has the potential to progress to serious infection. This highlights that intensified surveillance was needed for subsequent infections in patients with MDRP bacteriuria, especially in patients with ICU acquisition or recurrent bacteriuria episodes.

## 4. Materials and Methods

### 4.1. Study Design and Patient Population

We retrospectively reviewed the medical records of patients with MDRP bacteriuria admitted to Inje University Busan Paik Hospital, an 850-bed tertiary care teaching hospital in Busan, South Korea, between January 2009 and December 2018. The hospital has four ICUs with a total of 56 beds and a hematopoietic stem cell transplantation unit. The study protocol was approved by the Institutional Review Board of this hospital (IRB number: 2022-11-010-001), and informed consent was waived due to the retrospective nature of the analysis.

This study employed a retrospective cohort design. We identified a cohort of patients with MDRP bacteriuria at baseline and followed them retrospectively for 12 months to assess the occurrence of subsequent symptomatic MDRP infections and mortality. All patients aged ≥ 18 years with positive urine culture for MDRP bacteriuria during the study period were included. For patients with multiple episodes of MDRP bacteriuria during the study period, only the first episode for each patient was included in the analysis. Patients who had a symptomatic MDRP infection diagnosed prior to the identification of MDRP bacteriuria were excluded. To ensure accurate classification of subsequent symptomatic infections, we employed specific temporal criteria. For patients with asymptomatic bacteriuria, we confirmed the occurrence of subsequent symptomatic infections only if they developed at least 48 h after the initial diagnosis of bacteriuria. In cases of symptomatic bacteriuria, we identified subsequent symptomatic infections only if they occurred at least 48 h after the completion of the initial treatment regimen. The following demographic and clinical characteristics were collected: age, sex, underlying diseases, urinary tract catheter use during the bacteriuria episode, presence of other microbes in the urine, pyuria, and signs and symptoms suggestive of UTI according to IDSA guidelines [25,26]. ICU acquisition, healthcare exposure within 30 days prior to MDRP bacteriuria, and antimicrobial exposure within 90 days prior to MDRP bacteriuria were also included.

The primary endpoint was the incidence of subsequent symptomatic MDRP infections at any site within 12 months following the initial MDRP bacteriuria episode. MDRP infections with same susceptibility profiles to those of the original urine culture were included. The secondary endpoint was the overall mortality rate within 12 months in patients with MDRP bacteriuria.

### 4.2. Definitions

Bacteriuria was defined as a positive urine culture that grew ≥ 10,000 CFU/mL. MDRP was defined when it was not susceptible to one or more agents in at least three antimicrobial categories, whereas extensively drug-resistant *P. aeruginosa* was defined when it was not susceptible to at least one agent in all but two or fewer antimicrobial categories [27]. Hospital-acquired infection was defined as MDRP bacteriuria in a patient admitted for more than 48 h. Healthcare-associated community-onset infection was confirmed as MDRP bacteriuria when healthcare exposure, such as outpatient chemotherapy or dialysis, occurred within 30 days of onset. Symptomatic infections were defined based on clinical manifestation, laboratory results, and radiologic findings according to National Healthcare Safety Network surveillance definitions [28]. Gastrointestinal infections were categorized as gastrointestinal or hepatobiliary tract infections. Respiratory tract infections encompassed both non-ventilator-associated and ventilator-associated pneumonia. Active antimicrobial therapy was determined based on antibiotics that demonstrated in vitro activity against *P. aeruginosa* isolates during the treatment period.

### 4.3. Microbiological Methods

Bacterial identification and antimicrobial susceptibility tests were performed using the Vitek II automated system (bioMérieux, Hazelwood, MO, USA). The VITEK 2 Gram Negative Susceptibility Card (AST-N225) was used to determine the antimicrobial susceptibility. The results of the antimicrobial susceptibility tests were interpreted based on the CLSI guidelines [29]. Intermediate susceptibility was defined as being non-susceptible.

### 4.4. Statistical Analysis

All statistical analyses were performed using IBM SPSS Statistics, version 27.0 (IBM Corp., Armonk, NY, USA). The patient population was divided into two groups: those who subsequently developed symptomatic MDRP infection and those who remained without infection. To identify potential risk factors at the onset of bacteriuria which were associated with progression to symptomatic infection and overall mortality, bivariate analyses were performed. The χ2 test or Fisher’s exact test were used for categorical variables, and two-sample t-test or Mann–Whitney test variables were used for continuous variables. A multivariable logistic regression model was employed to determine independent risk factors for symptomatic infection and mortality. Variables with a *p* value < 0.05 in the bivariate analysis were included in the subsequent multivariable analysis. The Hosmer–Lemeshow statistic was used to assess the goodness of fit of the final model. *p*-values < 0.05 were considered statistically significant.

## 5. Conclusions

This study showed important insights into the long-term clinical course and outcomes of MDRP bacteriuria, highlighting the potential for progression to serious infections and associated mortality. Our findings could contribute to the development of targeted management strategies and emphasise the importance of prevention strategies to reduce subsequent infections in patients with MDRP bacteriuria.

## Figures and Tables

**Figure 1 antibiotics-13-00685-f001:**
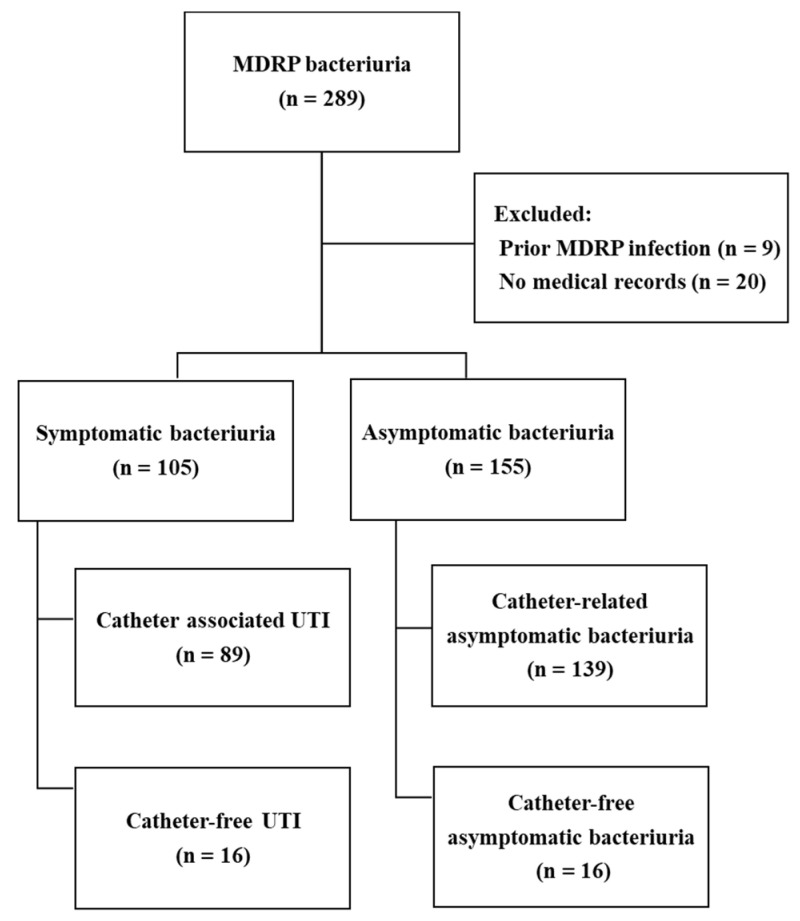
Flow diagram of patient selection process. MDRP, multidrug-resistant *Pseudomonas aeruginosa*; UTI, urinary tract infection.

**Table 1 antibiotics-13-00685-t001:** Characteristics of patients with MDRP bacteriuria for progression to symptomatic MDRP infection within 12 months.

Parameters at the Initial MDRP Bacteriuria Episode	Total (n = 260)	No Subsequent Symptomatic MDRP Infection (n = 181)	Subsequent Symptomatic MDRP Infection (n = 79)	*p*-Value
Median age, years (IQR)	70 (20–94)	70 (25–94)	69 (20–93)	0.10
Age ≥ 65 years	162 (62.3)	118 (65.2)	44 (55.7)	0.15
Male	170 (65.4)	119 (65.7)	51 (64.6)	0.85
Hospital-acquired	174 (66.9)	121 (66.9)	53 (67.1)	0.97
Healthcare-associated	86 (33.1)	60 (33.1)	26 (32.9)	-
Underlying diseases				
Diabetes mellitus	57 (21.9)	41 (22.7)	16 (20.3)	0.67
Solid tumour	48 (18.5)	37 (20.4)	11 (13.9)	0.21
Genitourinary malignancy	15 (5.8)	11 (6.1)	4 (5.1)	0.50
Cardiovascular disease	104 (40.0)	75 (41.4)	29 (36.7)	0.47
Chronic liver disease	14 (5.4)	10 (5.5)	4 (5.1)	0.57
Chronic lung disease	12 (4.6)	10 (5.5)	2 (2.5)	0.36
Chronic renal disease	34 (13.1)	22 (12.2)	12 (15.2)	0.50
Urological disease	55 (21.2)	31 (17.1)	24 (30.4)	0.02
Neurologic disease	184 (70.8)	125 (69.1)	59 (74.7)	0.36
ICU admission at diagnosis	21 (8.1)	11 (6.1)	10 (12.7)	0.04
Symptomatic bacteriuria	105 (40.4)	65 (35.9)	40 (50.6)	0.03
Asymptomatic bacteriuria	155 (59.6)	116 (64.1)	39 (49.4)	-
Microscopic pyuria	202 (78.0)	137 (75.7)	65 (82.3)	0.27
Catheter-associated bacteriuria	228 (87.7)	158 (87.3)	70 (88.6)	0.77
Concurrent bacteraemia during the initial bacteriuria episode	6 (2.3)	3 (1.7)	3 (3.8)	0.36
XDRP	191 (73.5)	129 (71.3)	62 (78.5)	0.23
Treatment				
Active antibiotic treatment for MDRP	70 (26.9)	41 (31.5)	29 (50.0)	0.02
Catheter outcome (n = 228)				
Removed within 7 days	38 (16.7)	33 (23.1)	5 (6.0)	<0.01
Exchanged within 7 days	136 (59.6)	85 (59.4)	51 (76.1)	
Remained more than 7 days	37 (16.2)	25 (17.5)	12 (17.9)	
Recurrent bacteriuria *	90 (34.7)	44 (24.4)	46 (58.2)	<0.01

IQR, interquartile range; ICU, intensive care unit; UTI, urinary tract infection; MDRP, multidrug-resistant *Pseudomonas aeruginosa*; XDRP, extensively drug-resistant *Pseudomonas aeruginosa*. Data are n (%) unless otherwise stated. * Recurrent bacteriuria was defined as the occurrence of two or more episodes of bacteriuria, either during the interval between the initial bacteriuria and the onset of symptomatic infection, or within the 12-month follow-up period from the initial bacteriuria episode.

**Table 2 antibiotics-13-00685-t002:** Multivariate analysis of risk factors for symptomatic MDRP infections within 12 months of MDRP bacteriuria.

Variables at the Initial MDRP Bacteriuria Episode	Odds Ratio	95% Confidence Interval	*p* Value	Adjusted Odds Ratio	95% Confidence Interval	*p* Value
Underlying urologic diseases (n = 55)	2.11	1.14–3.91	0.02			
ICU admission (n = 21)	2.24	1.91–5.51	0.04	4.12	1.23–13.88	0.02
Symptomatic bacteriuria (n = 105)	1.83	1.07–3.13	0.03			
Active antibiotic therapy (n = 70)	2.17	1.15–4.09	0.02			
Catheter removal within 7 days (n = 38)	0.28	0.11–0.76	0.01	0.26	0.07–1.05	0.06
Recurrent bacteriuria * (n = 90)	4.40	2.46–7.56	<0.01	4.24	1.88–9.38	<0.01

ICU, intensive care unit; MDRP, multidrug-resistant *Pseudomonas aeruginosa*. Variables with a *p*-value < 0.05 in the univariate analyses are included in the subsequent multivariate logistic regression model. Hosmer–Lemeshow test, χ2 = 10.676, *p* = 0.221. * Recurrent bacteriuria was defined as the occurrence of two or more episodes of bacteriuria, either during the interval between the initial bacteriuria and the onset of symptomatic infection or within the 12-month follow-up period from the initial bacteriuria episode.

**Table 3 antibiotics-13-00685-t003:** Comparison of characteristics between survivors and non-survivors with MDRP bacteriuria.

Parameters at the Initial MDRP Bacteriuria Episode	Survivor (n = 216)	Non-Survivor (n = 44)	*p*-Value
Median age, years (IQR)	69 (20–93)	72(44–94)	0.09
Age ≥ 65 years	131 (60.6)	31 (70.5)	0.22
Male	140 (64.8)	30 (68.2)	0.67
Underlying diseases			
Diabetes mellitus	43 (19.9)	14 (31.8)	0.08
Solid tumour	34 (15.7)	14 (31.8)	0.01
Cardiovascular disease	75 (34.7)	29 (65.9)	<0.01
Chronic liver disease	7 (3.2)	7 (15.9)	0.01
Chronic lung disease	3 (1.4)	9 (50.5)	<0.01
Chronic renal disease	25 (11.6)	9 (20.5)	0.11
Urological disease	48 (22.2)	7 (15.9)	0.35
Neurologic disease	155 (71.8)	29 (65.9)	0.44
ICU admission	14 (6.5)	7 (15.9)	0.04
Asymptomatic bacteriuria	127 (58.8)	28 (63.6)	0.55
Symptomatic bacteriuria	89 (41.2)	16 (36.4)	0.55
Microscopic pyuria	174 (80.6)	28 (65.1)	0.03
Catheter-associated bacteriuria	190 (88.0)	38 (86.4)	0.77
Concurrent bacteraemia during the initial bacteriuria episode	3 (1.4)	3 (6.8)	0.03
XDRP	159 (73.6)	32 (72.7)	0.90
Treatment			
Active antibiotic treatment for MDRP	57 (37.5)	13 (36.1)	0.88
Catheter outcome			
Removed within 7 days	33 (18.6)	4 (12.1)	0.18
Exchanged within 7 days	110 (62.1)	26 (78.8)	
Remained more than 7 days	34 (19.2)	3 (9.1)	
Recurrent bacteriuria *	83 (38.6)	7 (15.9)	<0.01
Presence of subsequent symptomatic MDRP infection	64 (29.6)	15 (34.1)	0.56

IQR, interquartile range; ICU, intensive care unit; UTI, urinary tract infection; MDRP, multidrug-resistant *Pseudomonas aeruginosa*; XDRP, extensively drug-resistant *Pseudomonas aeruginosa*. Data are n (%) unless otherwise stated. * Recurrent bacteriuria was defined as the occurrence of two or more episodes of bacteriuria, either during the interval between the initial bacteriuria and the onset of symptomatic infection or within the 12-month follow-up period from the initial bacteriuria episode.

**Table 4 antibiotics-13-00685-t004:** Multivariate analysis of risk factors for 12-month mortality in MDRP bacteriuria patients.

Variables at the Initial MDRP Bacteriuria Episode	Odds Ratio	95% Confidence Interval	*p*-Value	Adjusted Odds Ratio	95% Confidence Interval	*p*-Value
Solid tumour (n = 48)	2.50	1.20–5.20	0.01	2.92	1.18–7.27	0.02
Cardiovascular disease (n = 104)	3.64	1.84–7.20	<0.01	3.44	1.53–7.72	<0.01
Chronic liver disease (n = 14)	5.65	1.87–17.05	<0.01	6.57	1.84–23.48	<0.01
Chronic lung disease (n = 12)	18.25	4.71–71.76	<0.01	21.85	4.95–96.40	<0.01
ICU admission (n = 21)	2.73	1.03–7.22	0.04	5.11	1.53–17.08	<0.01
Absence of pyuria (n = 58)	2.22	1.86–4.55	0.03	2.50	1.06–5.89	0.04
Concurrent bacteraemia during the initial bacteriuria episode (n = 6)	5.20	1.01–26.64	0.03	7.34	1.16–46.42	0.03
Recurrent bacteriuria * (n = 90)	0.30	0.13–0.71	<0.01			

MDRP, multidrug-resistant *Pseudomonas aeruginosa*; ICU, intensive care unit. Variables with a *p*-value < 0.05 in the univariate analyses are included in the subsequent multivariate logistic regression model. Hosmer–Lemeshow test, χ2 = 1.649, *p* = 0.977. * Recurrent bacteriuria was defined as the occurrence of two or more episodes of bacteriuria, either during the interval between the initial bacteriuria and the onset of symptomatic infection or within the 12-month follow-up period from the initial bacteriuria episode.

## Data Availability

The datasets used during the current study are available from the corresponding author on reasonable request.

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
