# Peer review of "Long-Term Outcomes of Multidrug-Resistant Pseudomonas aeruginosa Bacteriuria: A Retrospective Cohort Study"

_antibiotics, 2024, doi:10.3390/antibiotics13080685_

Round 1

Reviewer 1 Report

Comments and Suggestions for Authors

Thank you very much for allowing me to review this work. This is a descriptive study that aims to illuminate potential predictors of symptomatic MDRP infections following prior bacteriuria by this pathogen, as well as mortality from MDRP bacteriuria. The study also provides a description of the prevalence of symptomatic and asymptomatic bacteriuria caused by MDRP, associated comorbidities, and sociodemographic variables of the patients. It is a well-conducted and systematic study, although I have several questions that I would like the authors to address:

  1. Abstract: Well-structured. In the results subsection, I do not understand why infections at locations other than the urinary tract are described. Initially, the study's sample selection criteria state that patients must have a diagnosis of bacteriuria caused by MDRP. So, why are MDRP infections from other locations described? Are they coinfections? Are they primary or secondary infections to the bacteriuria? This should be defined and explained. It is also reported in this section and the results section that 31.8% of the overall mortality in the sample is attributed to MDRP infection. In this type of study, a descriptive cross-sectional study, it is risky to discuss attributable mortality without prospective patient follow-up, as calculating attributable risk is not possible with this design. The terminology used in the results should be more precise.

  2. Introduction: The introduction should focus more on MDRP bacteriuria. For example, line 33 mentions ventilator-associated pneumonia, which is not the focus of the current study. The introduction is well-structured, short, and easy to read and understand, but it should concentrate solely on MDRP bacteriuria.

  3. Results: a. In line 66, an acronym, ASB, appears without definition, neither in that line nor previously. It should be defined. b. The same issue occurs in line 72 with the acronym XDRP, which is not defined in that line or previously. It is defined later in the tables, but the acronym should be defined the first time it appears in the text. c. The representation of p-values should use a lowercase 'p'. Additionally, they should be presented with three decimal places, not four as in the text. For p-values less than 0.01, they should be represented as p<0.01. d. Table 1: Columns 3 and 4 are labeled "absence of symptomatic infection" and "symptomatic MDRP infection," respectively. Why are they defined this way? Why compare symptomatic infections (without knowing if they are caused by MDRP or other pathogens) with symptomatic MDRP infections? Logically, based on the main objective, it would be more appropriate to compare the symptomatic MDRP bacteriuria group with the asymptomatic MDRP bacteriuria group. e. In Tables 2 and 4, it would be beneficial to present crude and adjusted OR values for the different variables that could act as confounders. This would allow readers to better estimate the presented OR values. f. The OR intervals presented in Tables 2 and 4 are wide. The value of the ORs with these 95% CIs should be discussed. The tables should also include the analyzed sample size (n).

  4. Discussion: In this section (and others throughout the text), "incidence" is mentioned. This terminology should be revised. This study does not allow for the determination of cumulative incidence or incidence rate (incidence density). Only prevalence can be estimated. Please correct this.

  5. Materials and Methods: The design should be correctly reported: this study is a descriptive cross-sectional study with historical data collection, not a retrospective design.

Thank you very much.

Author Response

Dear Sir

July 19, 2024

I really appreciate all reviewers for critical and helpful suggestions, and I feel that the quality of the manuscript has been significantly improved as a result. I provide point-by-point responses to the reviewers' comments. The text in bold signifies the comments made by a reviewer. The authors’ responses appear below each comment.

Modified portions were highlighted in yellow in the manuscript.

-------------------------------------------------------------------------------------------------------------

Reviewer 1: Thank you very much for allowing me to review this work. This is a descriptive study that aims to illuminate potential predictors of symptomatic MDRP infections following prior bacteriuria by this pathogen, as well as mortality from MDRP bacteriuria. The study also provides a description of the prevalence of symptomatic and asymptomatic bacteriuria caused by MDRP, associated comorbidities, and sociodemographic variables of the patients. It is a well-conducted and systematic study, although I have several questions that I would like the authors to address:

5. Materials and Methods: The design should be correctly reported: this study is a descriptive cross-sectional study with historical data collection, not a retrospective design.

We appreciate your thoughtful evaluation of our study design. While we understand the perspective that led to classifying this as a cross-sectional study, we respectfully submit that our methodology aligns more closely with a retrospective cohort study design. Our rationale for this classification is as follows:

  • Temporal sequence: Our study examines outcomes (subsequent symptomatic infections and mortality) that occur after the initial exposure (MDRP bacteriuria) over a defined follow-up period of 12 months.
  • Cohort definition: We identified a specific cohort (patients with MDRP bacteriuria) at baseline and followed this cohort retrospectively to assess outcomes.
  • Incidence measurement: Our design allows for the calculation of incidence rates of subsequent infections and mortality, which is characteristic of cohort studies rather than cross-sectional studies.

We acknowledge that the retrospective nature of data collection may have led to the initial impression of a cross-sectional design. However, the longitudinal aspect of our outcome assessment over a 12-month period distinguishes this study from a true cross-sectional design, which typically provides a snapshot at a single point in time.

We hope this explanation clarifies our methodological approach. We are grateful for your insight, as it has prompted us to provide a more detailed description of our study design in the manuscript as follows.

Line 14. A retrospective cohort study was conducted among patients with MDRP bacteriuria who hospitalized at a tertiary care hospital from 2009 to 2018, with a 12-month follow-up period for each patient. The primary endpoint was the incidence of subsequent symptomatic MDRP infections at any site and the secondary endpoint was the overall mortality rate.

Line 237. This study employed a retrospective cohort design. We identified a cohort of patients with MDRP bacteriuria at baseline and followed them retrospectively for 12 months to assess the occurrence of subsequent symptomatic MDRP infections and mortality.

1. Abstract: Well-structured. In the results subsection, I do not understand why infections at locations other than the urinary tract are described. Initially, the study's sample selection criteria state that patients must have a diagnosis of bacteriuria caused by MDRP. So, why are MDRP infections from other locations described? Are they coinfections? Are they primary or secondary infections to the bacteriuria? This should be defined and explained. It is also reported in this section and the results section that 31.8% of the overall mortality in the sample is attributed to MDRP infection. In this type of study, a descriptive cross-sectional study, it is risky to discuss attributable mortality without prospective patient follow-up, as calculating attributable risk is not possible with this design. The terminology used in the results should be more precise.

The infections described at various body sites are not coinfections, but rather subsequent infections that developed following the initial MDRP bacteriuria. Our study aimed to investigate the potential progression of MDRP bacteriuria to symptomatic infections at any anatomical site. We have revised the abstract to clarify this temporal relationship as follows.

Line 19. Subsequent symptomatic MDRP infections were documented in 79 patients (30.3%) within 12 months of the initial bacteriuria episode.

Upon careful reflection, we concur that our study design is more accurately classified as a retrospective cohort study rather than a cross-sectional study. This is because we followed a defined cohort of patients with MDRP bacteriuria over time to assess subsequent outcomes.

Regarding attributable mortality, we acknowledge the limitations of our study design in establishing direct causality. To address this, we have revised our language to more accurately reflect our findings as follows.

Line 24. The overall mortality rate was 16.9%, with 31.8% of deaths estimated to be associated with MDRP infection.

2. Introduction: The introduction should focus more on MDRP bacteriuria. For example, line 33 mentions ventilator-associated pneumonia, which is not the focus of the current study. The introduction is well-structured, short, and easy to read and understand, but it should concentrate solely on MDRP bacteriuria.

We thank the reviewer for this suggestion. We have refocused the introduction to concentrate specifically on MDRP bacteriuria as follows.

Line 35. Urinary tract infections (UTIs) represent the most prevalent form of nosocomial infections [10], with a growing concern over the increasing incidence of antibiotic-resistant gram-negative pathogens in recent years. Among these, Pseudomonas aeruginosa is responsible for a significant proportion of UTI cases, accounting for 7-10% of infections [12]. Of particular concern is the rise of multidrug-resistant P. aeruginosa (MDRP) strains, which have become increasingly prevalent in hospital settings [13-15]. MDRP poses a significant challenge in healthcare environments due to its ability to transmit within hospitals, potentially triggering outbreaks [3-5]. This characteristic, combined with its resistance to multiple antibiotics, complicates treatment strategies and raises concerns about patient outcomes. Although some antibiotics have shown efficacy against MDRP, optimal treatment approaches remain unclear, necessitating a delicate balance between judicious antimicrobial use and the risks associated with disseminated multidrug-resistant infections [7-9].

Bacteriuria often leads to unnecessary antimicrobial use and urinary drainage systems can serve as reservoirs and potential sources of multidrug-resistant bacteria transmission to other patients [40,41]. The relationship between bacteriuria and subsequent symptomatic infections, particularly bacteremia, has been a subject of ongoing research. Previous studies have identified several risk factors for bacteremia originating from urinary sources, including diabetes, immunosuppression, catheterization, and the presence of shaking chills [42-44]. However, studies investigating the factors associated with sub-sequent symptomatic infection in patients with MDRP bacteriuria are lacking. Moreover, the long-term outcomes and mortality of patients with MDRP bacteriuria have not yet been defined. Therefore, our study aimed to investigate the clinical characteristics, long-term outcomes, and factors associated with symptomatic infection and mortality among patients with MDRP bacteriuria, and to develop a management strategy for this condition.

3. Results: 

a. In line 66, an acronym, ASB, appears without definition, neither in that line nor previously. It should be defined.

We have defined the acronym ASB (asymptomatic bacteriuria) upon first use as follows.

Line 67. 59.6% (n=155), 34.2% (n=89), and 6.2% (n=16) were classified as having asymptomatic bacteriuria (ASB), catheter-related UTI, and catheter-free UTI, respectively

b. The same issue occurs in line 72 with the acronym XDRP, which is not defined in that line or previously. It is defined later in the tables, but the acronym should be defined the first time it appears in the text.

We have defined XDRP (extensively drug-resistant Pseudomonas) upon first appearance in the text.

Line 74. MDRP and extensively drug-resistant Pseudomonas (XDRP), respectively.

c. The representation of p-values should use a lowercase 'p'. Additionally, they should be presented with three decimal places, not four as in the text. For p-values less than 0.01, they should be represented as p<0.01.

We have corrected the p-value formatting as suggested.

d. Table 1: Columns 3 and 4 are labeled "absence of symptomatic infection" and "symptomatic MDRP infection," respectively. Why are they defined this way? Why compare symptomatic infections (without knowing if they are caused by MDRP or other pathogens) with symptomatic MDRP infections? Logically, based on the main objective, it would be more appropriate to compare the symptomatic MDRP bacteriuria group with the asymptomatic MDRP bacteriuria group.

We understand the confusion this may have caused and would like to clarify our approach and rationale. The primary objective of our study was to investigate factors associated with subsequent symptomatic infection in patients with initial MDRP bacteriuria, as well as to examine mortality in this patient population. Our focus was on the progression from initial MDRP bacteriuria to subsequent symptomatic MDRP infection. To address your concerns and improve clarity, we have made the following changes: We have relabeled Table 1 column 4 as "Subsequent symptomatic MDRP infection" instead of "symptomatic MDRP infection. We also changed Table 1 column 1 “parameters” into “parameters at the initial MDRP bacteriuria episode.

Table 1. Characteristics of patients with MDRP bacteriuria for progression to symptomatic MDRP infection within 12 months.

Parameters at the initial MDRP bacteriuria episode

Total

(n=260)

No subsequent symptomatic MDRP infection

(n=181)

Subsequent symptomatic MDRP infection

(n=79)

p value

e. In Tables 2 and 4, it would be beneficial to present crude and adjusted OR values for the different variables that could act as confounders. This would allow readers to better estimate the presented OR values.

We have included crude and adjusted OR values in Tables 2 and 4.

Table 2. Multivariate analysis of risk factors for symptomatic MDRP infections within 12 months of MDRP bacteriuria.

Variables at the initial MDRP bacteriuria episode

OR

95% CI

p value

Adjusted OR

95% CI

p value

Undelying urologic diseases (n=55)

2.11

1.14-3.91

0.02

1.09

0.44-2.67

0.85

ICU admission (n=21)

2.24

1.91-5.51

0.04

4.12

1.23-13.88

0.02

Symptomatic bacteriuria (n=105)

1.83

1.07-3.13

0.03

1.99

0.86-4.60

0.11

Active antibiotic therapy (n=70)

2.17

1.15-4.09

0.02

1.76

0.79-3.94

0.17

Recurrent bacteriuria* (n=90)

4.40

2.46-7.56

<0.01

4.24

1.88-9.38

<0.01

Catheter removal within 7 days (n=38)

0.28

0.11-0.76

0.01

0.26

0.07-1.05

0.06

Table 4. Multivariate analysis of risk factors for 12-months mortality in MDRP bacteriuria patients.

Variables at the initial MDRP bacteriuria episode

OR

95% CI

P-value

Adjusted OR

95% CI

P-value

Solid tumor (n=48)

2.50

1.20-5.20

0.01

2.92

1.18-7.27

0.02

Cardiovascular disease (n=104)

3.64

1.84-7.20

<0.01

3.44

1.53-7.72

<0.01

Chronic liver disease (n=14)

5.65

1.87-17.05

<0.01

6.57

1.84-23.48

<0.01

Chronic lung disease (n=12)

18.25

4.71-71.76

<0.01

21.85

4.95-96.40

<0.01

ICU admission (n=21)

2.73

1.03-7.22

0.04

5.11

1.53-17.08

<0.01

Absence of pyuria (n=58)

2.22

1.86-4.55

0.03

2.50

1.06-5.89

0.04

Concurrent bacteraemia during the initial bacteriuria episode (n=6)

5.20

1.01-26.64

0.03

7.34

1.16-46.42

0.03

Recurrent bacteriuria* (n=90)

0.30

0.13-0.71

<0.01

f. The OR intervals presented in Tables 2 and 4 are wide. The value of the ORs with these 95% CIs should be discussed. The tables should also include the analyzed sample size (n).

The wide 95% CIs observed in our study, despite statistically significant p-values (p < 0.05), indeed suggest a degree of uncertainty in our point estimates. This width can be attributed to several factors, primarily the relatively small number of events for certain predictor variables. For instance, in Table 2, ICU admission was identified as an independent predictor, but the total number of patients admitted to the ICU was only 21. Similarly, in Table 4, we observed low event rates for several key variables: 21 patients with ICU admission, 12 patients with chronic liver disease, 12 patients with chronic lung disease, and 6 patients with concurrent bacteraemia during the initial bacteriuria episode. These low event rates substantially contribute to the width of the CIs for these predictors. The limited number of events for these variables, while sufficient to detect statistically significant associations, results in less precise estimates of the magnitude of these associations. This is reflected in the wide CIs, which indicate a broader range of plausible values for the true population parameter.

To address these limitations, we have added the following paragraph to our discussion:

Line 214. "While our findings identify several significant predictors of subsequent symptomatic infection and mortality, the wide confidence intervals, particularly for variables with low event rates, suggest a degree of uncertainty in the magnitude of these associations. For instance, the small number of patients with ICU admission (n=21), chronic liver disease (n=14), chronic lung disease (n=12), and concurrent bacteraemia (n=6) contribute to the imprecision of these estimates. These results should be interpreted as indicative of potential risk factors that warrant further investigation in larger, prospectively designed studies with a priori sample size calculations to ensure adequate statistical power for less common predictor variables."

4. Discussion: In this section (and others throughout the text), "incidence" is mentioned. This terminology should be revised. This study does not allow for the determination of cumulative incidence or incidence rate (incidence density). Only prevalence can be estimated. Please correct this.

We believe the use of 'incidence' is appropriate and accurately reflects the nature of our data and analyses. However, we acknowledge that the distinction between retrospective cohort studies and cross-sectional studies can sometimes be subtle, and we appreciate the reviewer's attention to this important methodological point.

---------------------------------------------------------------------------------------------------------------------------------------

We hope that these revisions adequately address the reviewers' concerns. We are grateful for their valuable input, which has helped to enhance the scientific rigor and clarity of our manuscript. We look forward to your feedback on these changes.

Sincerely,

Yu Mi Wi

Reviewer 2 Report

Comments and Suggestions for Authors

Very good article. In retrospective studies, the authors examined the long-term outcomes of Pseudomonas aeruginosa bacteriuria. The paper presents detailed results regarding patients, risk factors, comparisons of characteristics between survivors and non-survivors.

I believe that the article is of great clinical importance and is worth publishing with a few corrections:

1. Add patient inclusion and exclusion criteria.

2. Figure 1 is small and difficult to read.

3. Why was the CLSI interpretation used? CLSI is mainly used in environmental, biological and agricultural microbiology. For medical and clinical research, as in this article, the EUCAST interpretation is intended https://www.eucast.org/ Therefore, the authors must use the EUCAST interpretation and partially change their results.

4. When describing the Vitek II methodology, please provide what casettes were used.

Comments on the Quality of English Language

OK

Author Response

Dear Sir

July 19, 2024

I really appreciate all reviewers for critical and helpful suggestions, and I feel that the quality of the manuscript has been significantly improved as a result. I provide point-by-point responses to the reviewers' comments. The text in bold signifies the comments made by a reviewer. The authors’ responses appear below each comment.

Modified portions were highlighted in yellow in the manuscript.

-------------------------------------------------------------------------------------------------------------

Reviewer 2:

Very good article. In retrospective studies, the authors examined the long-term outcomes of Pseudomonas aeruginosa bacteriuria. The paper presents detailed results regarding patients, risk factors, comparisons of characteristics between survivors and non-survivors.

I believe that the article is of great clinical importance and is worth publishing with a few corrections:

1. Add patient inclusion and exclusion criteria.

We have added a detailed description of patient inclusion and exclusion criteria to the Methods section as follows.

Line 239. All patients aged ≥18 years with positive urine culture for MDRP bacteriuria during the study period were included. For patients with multiple episodes of MDRP bacteriuria during the study period, only the first episode for each patient was included in the analysis. Patients who had a symptomatic MDRP infection diagnosed prior to the identification of MDRP bacteriuria were excluded. To ensure accurate classification of subsequent symptomatic infections, we employed specific temporal criteria. For patients with asymptomatic bacteriuria, we confirmed the occurrence of subsequent symptomatic infections only if they developed at least 48 hours after the initial diagnosis of bacteriuria. In cases of symptomatic bacteriuria, we identified subsequent symptomatic infections only if they occurred at least 48 hours after the completion of the initial treatment regimen.

2. Figure 1 is small and difficult to read.

We apologize for the poor quality of Figure 1. We have recreated the figure at a higher resolution to improve readability.

3. Why was the CLSI interpretation used? CLSI is mainly used in environmental, biological and agricultural microbiology. For medical and clinical research, as in this article, the EUCAST interpretation is intended https://www.eucast.org/ Therefore, the authors must use the EUCAST interpretation and partially change their results.

We appreciate the reviewer's insightful comment regarding the use of CLSI versus EUCAST interpretations. While we acknowledge the growing adoption of EUCAST guidelines in clinical research and the potential impact of differing breakpoints on result interpretation, our study's context necessitated the use of CLSI standards. In South Korea, where this study was conducted, clinical microbiology laboratories predominantly utilize CLSI-based automated systems for susceptibility testing. Given the retrospective nature of our study, we were constrained to the data available from the time of patient care, which aligned with CLSI interpretations. For future prospective studies, we commit to incorporating both CLSI and EUCAST interpretations to provide a more comprehensive analysis of antimicrobial susceptibility patterns and their clinical implications. We are grateful for this valuable feedback, which will undoubtedly enhance the robustness of our future research in this field.

4. When describing the Vitek II methodology, please provide what casettes were used.

We have provided details on the specific Vitek II cassettes used in our methodology as follows.

Line 281. The VITEK 2 Gram Negative Susceptibility Card (AST-N225) was used to determine the antimicrobial susceptibility.

---------------------------------------------------------------------------------------------------------------------------------------

We hope that these revisions adequately address the reviewers' concerns. We are grateful for their valuable input, which has helped to enhance the scientific rigor and clarity of our manuscript. We look forward to your feedback on these changes.

Sincerely,

Yu Mi Wi

Reviewer 3 Report

Comments and Suggestions for Authors

Line 23: ICU length of stay would be more appropriate risk factors than just admission. 

Line 54: I am not sure the statement about factors associated with symptomatic infection.In my opinion and also literature suggest that factors are well established contrary to what the authors are suggesting. 

Table 2 and 4 .Can the authors explicitly state the method used. Multivariate is too general a term. 

Under the Discussion 

Line 138: Sentence is incomplete. 

Line 153: The sentence is incomplete 

Line 166: How long were patients shedding the pathogen? Intermittently or continously?

Statistical Analysis 

The variables that were presented for analysing should be stated here and why you think they are important. 

The methods used must be justified why you think they were appropriate. 

My understanding is that if you had considered the multiple episodes for each individual was diagnosed with the infection, the Poisson Regression Modelling might have been more appropriate for your datasets than the logistic regression. 

Also, can you explain in detail how you examined model fit. Did you examine residuals etc? Which variables made the final result etc. This information is important. 

For example, Table 4 doesn’t show any standard errors. Those are very important parameters when assessing model performance.  Please  show us all the important aspects not just the p-values.

Author Response

Dear Sir

July 19, 2024

I really appreciate all reviewers for critical and helpful suggestions, and I feel that the quality of the manuscript has been significantly improved as a result. I provide point-by-point responses to the reviewers' comments. The text in bold signifies the comments made by a reviewer. The authors’ responses appear below each comment.

Modified portions were highlighted in yellow in the manuscript.

-------------------------------------------------------------------------------------------------------------

Reviewer 3:

Line 23: ICU length of stay would be more appropriate risk factors than just admission.

We appreciate this insightful suggestion. We agree that ICU length of stay could indeed be a more nuanced risk factor than ICU admission alone. In our future work, we will incorporate ICU length of stay as a continuous variable in our analyses to provide a more detailed assessment of ICU-related risk.

Line 54: I am not sure the statement about factors associated with symptomatic infection. In my opinion and also literature suggest that factors are well established contrary to what the authors are suggesting.

We thank the reviewer for this observation. We revised the part of introduction to more accurately reflect the existing literature on factors associated with subsequent symptomatic infection, while still highlighting the specific contributions of our study to this body of knowledge as follows.

Line 35. Urinary tract infections (UTIs) represent the most prevalent form of nosocomial infections [1], with a growing concern over the increasing incidence of antibiotic-resistant gram-negative pathogens in recent years. Among these, Pseudomonas aeruginosa is responsible for a significant proportion of UTI cases, accounting for 7-10% of infections [2]. Of particular concern is the rise of multidrug-resistant P. aeruginosa (MDRP) strains, which have become increasingly prevalent in hospital settings [3-5]. MDRP poses a significant challenge in healthcare environments due to its ability to transmit within hospitals, potentially triggering outbreaks [6-8]. This characteristic, combined with its resistance to multiple antibiotics, complicates treatment strategies and raises concerns about patient outcomes. Although some antibiotics have shown efficacy against MDRP, optimal treatment approaches remain unclear, necessitating a delicate balance between judicious antimicrobial use and the risks associated with disseminated multidrug-resistant infections [9-11].

Bacteriuria often leads to unnecessary antimicrobial use and urinary drainage systems can serve as reservoirs and potential sources of multidrug-resistant bacteria transmission to other patients [12,13]. The relationship between bacteriuria and subsequent symptomatic infections, particularly bacteremia, has been a subject of ongoing research. Previous studies have identified several risk factors for bacteremia originating from urinary sources, including diabetes, immunosuppression, catheterization, and the presence of shaking chills [14-16]. However, studies investigating the factors associated with subsequent symptomatic infection in patients with MDRP bacteriuria are lacking.

Table 2 and 4 .Can the authors explicitly state the method used. Multivariate is too general a term.

We apologize for the lack of specificity in our statistical methodology. We used multiple logistic regression for these analyses. We clearly state this as follows.

Line 111. Variables with a p-value < 0.05 in the univariate analyses are included in the subsequent multivariate logistic regression model. Hosmer–Lemeshow test, χ2 =10.676, p =0.221.

Line 135. Variables with a p-value < 0.05 in the univariate analyses are included in the subsequent multivariate logistic regression model. Hosmer–Lemeshow test, χ2 =1.649, p =0.977.

Under the Discussion 

Line 138/153: Sentence is incomplete.

We sincerely apologize for these oversights. We will carefully review and complete these sentences

Line 153. The incidence of bacteraemia in our study was 3.5%, which falls within the range of 0.4% to 4% reported in previous studies for patients with catheter-related bacteriuria who progress to bacteraemia.

Line 168. Notably, our study revealed that the incidence of subsequent infections was lower among patients whose catheters were removed compared to those whose catheters were ex-changed or maintained.

Line 166: How long were patients shedding the pathogen? Intermittently or continously?

Thank you for this important question. Our study did not specifically measure the duration or pattern of pathogen shedding. We acknowledge this as a limitation of our work and will add a statement in our discussion noting the need for future studies to examine the temporal dynamics of MDRP shedding in these patients.

Line 209. Third, our study included the lack of data on the duration and pattern of MDRP shedding in patients with bacteriuria. The temporal dynamics of pathogen shedding could potentially influence the risk of subsequent infections and transmission events. This information could provide valuable insights into the optimal duration of infection control measures and inform decisions about repeat cultures in the clinical management of MDRP bacteriuria.

Statistical Analysis 

The variables that were presented for analysing should be stated here and why you think they are important. The methods used must be justified why you think they were appropriate. My understanding is that if you had considered the multiple episodes for each individual was diagnosed with the infection, the Poisson Regression Modelling might have been more appropriate for your datasets than the logistic regression. Also, can you explain in detail how you examined model fit. Did you examine residuals etc? Which variables made the final result etc. This information is important. For example, Table 4 doesn’t show any standard errors. Those are very important parameters when assessing model performance. Please show us all the important aspects not just the p-values.

We appreciate the reviewer's insightful comments regarding our statistical methodology. We will address each point in turn:

Variable selection and importance:

Our variable selection was guided by both clinical relevance and previous literature. The relationship between bacteriuria and subsequent symptomatic infections, particularly bacteremia, has been a subject of ongoing research. Previous studies have identified several risk factors for bacteremia originating from urinary sources, including diabetes, immunosuppression, catheterization, and the presence of shaking chills [14-16]. However, studies investigating the factors associated with subsequent symptomatic infection specifically in patients with MDRP bacteriuria are lacking.

We considered variables identified in related studies, such as those by Conway et al. [17], Bursle et al. [18], and Advani et al. [19], which highlighted factors like age, sex, comorbidities, catheter use, and various clinical parameters. Through comprehensive chart review, we aimed to capture as much relevant patient information as possible to identify potential risk factors.

Choice of statistical method:

Our choice of logistic regression was based on our primary and secondary outcomes being binary (presence/absence of symptomatic infection and mortality). We did not consider multiple infection episodes per patient, which justifies our use of logistic regression over Poisson regression. We acknowledge that if we had included multiple episodes per patient, Poisson regression might have been more appropriate.

Model fit and variable selection:

To assess model fit, we used the Hosmer-Lemeshow goodness-of-fit test.

We employed a stepwise variable selection process, starting with univariate analyses to identify potentially significant variables (p < 0.05), which were then included in the multivariate model. Variables were retained in the final model based on their statistical significance and clinical relevance.

---------------------------------------------------------------------------------------------------------------

We hope that these revisions adequately address the reviewers' concerns. We are grateful for their valuable input, which has helped to enhance the scientific rigor and clarity of our manuscript. We look forward to your feedback on these changes.

Sincerely,

Yu Mi Wi

Round 2

Reviewer 1 Report

Comments and Suggestions for Authors

Thank you very much for addressing my concerns. You have clarified all my doubts perfectly. I believe the article has now improved in quality. Many thanks.

Reviewer 2 Report

Comments and Suggestions for Authors

The authors significantly corrected the manuscript according to the reviewer's suggestions. Recently, I recommend the article for publication.

Reviewer 3 Report

Comments and Suggestions for Authors

Thank you for addressing the my concerns.